# Preparation of Single-Helical Curdlan Hydrogel and Its Activation with Coagulation Factor G

**DOI:** 10.3390/polym16101323

**Published:** 2024-05-08

**Authors:** Geying Ru, Xiaoshuang Yan, Huijuan Wang, Jiwen Feng

**Affiliations:** 1Key Laboratory of Magnetic Resonance in Biological Systems, State Key Laboratory of Magnetic Resonance and Atomic and Molecular Physics, National Center for Magnetic Resonance in Wuhan, Wuhan Institute of Physics and Mathematics, Innovation Academy for Precision Measurement Science and Technology, Chinese Academy of Sciences, Wuhan 430071, China; 2University of Chinese Academy of Sciences, Beijing 100049, China

**Keywords:** β-1,3-glucan, curdlan, hydrogel, conformation, structure–activity relationship

## Abstract

β-1,3-glucans are a kind of natural polysaccharide with immunomodulatory, antitumor, and anti-inflammatory properties. Curdlan, as the simplest linear β-1,3-glucan, possesses a variety of biological activities and thermogelation properties. However, due to the complexity and variability of the conformations of curdlan, the exact structure–activity relationship remains unclear. We prepare a chemically crosslinked curdlan hydrogel with the unique single-helical skeleton (named S gel) in 0.4 wt% NaOH at 40 °C, confirmed by diffuse reflectance infrared Fourier transform spectroscopy (DRIFTS). X-ray diffractometry (XRD) data show that S gel maintains the single-helical crystal structure, and the degree of crystallinity of the S gel is ~24%, which is slightly lower than that of the raw powder (~31%). Scanning electron microscopy (SEM) reveals that S gel has a continuous network structure, with large pores measuring 50–200 μm, which is consistent with its high swelling property. Using the ^13^C high-resolution magic angle spinning nuclear magnetic resonance (HRMAS NMR) method, we determine that most of the single-helical skeleton carbon signals in the swollen S gel are visible, suggesting that the single-helical skeleton of S gel exhibits fascinating mobility at room temperature. Finally, we reveal that the binding of S gel to coagulation Factor G from tachypleus amebocyte lysate increases and saturates at 20 μL tachypleus amebocyte lysate per mg of S gel. Our prepared S gel can avoid the transformation of curdlan conformations and retain the bioactivity of binding to coagulation Factor G, making it a valuable material for use in the food industry and the pharmaceutical field. This work deepens the understanding of the relationship between the single-helical structure and the activity of curdlan, promoting the development and application of β-1,3-glucans.

## 1. Introduction

β-1,3-glucans are widely distributed in fungi, bacteria, plants, and algae, playing vital immunomodulatory, antitumor, and anti-inflammatory roles [1,2,3]. β-1,3-glucans interact not only with cell-surface receptors, such as dectin-1, complement receptor 3 (CR3), inhibitory surface glycoprotein CD5, and lactosylceramide (LacCer), but also with polynucleotides to produce composite structures of ordered geometries [4]. Curdlan is a non-branching linear β-1,3-glucan produced from the soil microorganism *Alcaligens faecalis* var. *10C3K*, which was first isolated and named by Harada et al. to describe its gelling behavior at elevated temperatures [5]. The thermal gel characteristic of curdlan has attracted significant attention. For example, due to its colorless, tasteless, and non-digestible properties, curdlan gels can be used as a fat-mimicking substitute [6]. Curdlan also has a variety of biofunctions, such as an inhibitory effect against AIDS virus infection and mediating the coagulant pathway by activating Factor G (the first protease zymogen of the alternative coagulation pathway in the amebocyte lysate) [7,8].

The structure of curdlan is considered to be an important factor contributing to its biological activities. Curdlan exhibits three conformations—single-helix, triple-helix, and coil. These three conformations can undergo transformation when external environments or solution conditions change. Generally, thermo-reversible curdlan gel is prepared in the temperature range from 60 to 80 °C, while thermo-reversible curdlan gel is transformed into thermo-irreversible curdlan gel at a temperature between 80 and 120 °C [9,10]. The predominant opinion is that thermo-reversible curdlan gels form mainly single-helical structures, while thermo-irreversible curdlan gels form mainly triple-helical structures [11,12,13]. Using attenuated total reflectance infrared spectroscopy (ATR/FT-IR) to characterize the in situ gelation process of curdlan in water, Gagnon et al. suggested that the structure of thermo-reversible gels is an intermediate in the formation of the triple helix conformation [14]. Using atomic force microscopy (AFM) and transmission electron microscope (TEM) to visually observe curdlan morphologies at heating temperatures, it was noted that bundled curdlan triple helixes hydrated and swelled at 40 °C, separated from each other at temperatures from 25 to 50 °C, dissociated into partially opened triple-helical chains and single-helixes at 60 and 70 °C, and finally further dissociated into single-helixes above 80 °C [15]. In addition to breaking the triple helical structure of curdlan at temperatures above 80 °C, at a high pH (>0.24 M sodium hydrate (NaOH)) and in a dimethyl sulfoxide (DMSO) solution, the curdlan chain can be completely transformed into a random coil conformation [16]. Among these methods, the addition of a strong base (e.g., NaOH) is a convenient way to break the H-bonding at room temperature, which can be reversed by neutralization. Subsequent neutralization can further change the conformation of curdlan to a single helix.

It is widely accepted that the activities of β-1,3-glucans are related to their helical conformations [17]. Curdlan can exhibit a 100-fold increase in its ability to activate Factor G after alkali treatment, which should be associated with the complete or partial conversion from the triple-helical to the single-helical conformation [13]. That is to say, the single-helical conformation of β-1,3-glucan is a better stimulant for Factor G than is the triple-helical conformation [7]. The antitumor activity of curdlan also exhibited such a single-helix-specific response, although this specificity was less pronounced than that of branched β-1,3-glucans. Currently, there are three barriers to the biological applications of curdlan. (1) Curdlan is insoluble in water at ambient temperature, so treatments such as the addition of a strong base or DMSO can change the conformation of curdlan, and these high concentrations of NaOH or DMSO solvents are likely to affect living cells when injected in combination with curdlan, which complicate the interpretation of studies evaluating structure–activity relationships. (2) With the exception of our recently reported method of preparing the unique curdlan with the single-helical conformation in solution, complex and easy variable conformations of thermal curdlan hydrogel cannot give a clear picture of higher order structure, and thus, the separation of effects of higher order structures allowing their independent study remains a challenge. (3) As controlled drug delivery vehicles, biological materials like proteins may be destroyed under the conventional gel-forming conditions of curdlan (e.g., heating above 55 °C or treatment by alkaline).

Therefore, it is desirable to form curdlan gels with unique single- or triple-helical structures and minimization of chemical reactions. It is well known that curdlan chain is a random coil under high NaOH concentrations, and that it can transform into an ordered structure when the NaOH concentration decreases, although the helical structure is ambiguous [18,19]. Recently, our group found that curdlan fully maintains its single-helical conformation at NaOH concentrations below 0.6 wt% and temperatures ranging from 35 °C to 50 °C [20]. In order to understand the relationship between the single-helical structure and function of curdlan gel, we prepare a new kind of chemically crosslinked curdlan hydrogel with a unique single-helical skeleton (S gel). Then the structure, mobility, and activity of S gel in relation to Factor G are measured by diffuse reflectance infrared Fourier transform spectroscopy (DRIFTS), X-ray diffractometry (XRD), scanning electron microscopy (SEM), nuclear magnetic resonance spectroscopy (NMR) and ultraviolet–visible (UV–Vis) spectrophotometry.

## 2. Materials and Methods

### 2.1. Materials

Curdlan was purchased from Sigma-Aldrich (St. Louis, MI, USA), and the viscosity-average molecular weight, obtained using the gel permeation chromatography (GPC) method, was 7.9 × 10^5^ g/mol. Ethylene glycol diglycidyl ether (EGDGE), NaOH (99.99%), hydrochloric acid, and deuteroxide (D_2_O, 99.8% D), Tris•HCl buffer (pH 8.0) were purchased from J&K (Shanghai, China) and used without further treatment. Chromogenic end-point Tachypleus Amebocyte Lysate Kit (CE TAL Kit) was purchased from Xiamen Bioendo Technology Co., Ltd. (Fujian, China).

### 2.2. Synthesis of Curdlan Hydrogels with a Single-Helical Skeleton and a Random Skeleton

The preparation procedure of curdlan hydrogel with a single-helical skeleton (S gel) is as follows. Raw curdlan powder (1.0 g, 5 wt%) was dissolved in 0.4 wt% NaOH aqueous solution (20 mL) at 40 °C, then the excessive crosslinker EGDGE (10 mL) was added dropwise, under stirring, for 10 min. After 1 day, S gel was formed, and hydrochloric acid was added to neutralize NaOH. Then, the S gel was washed three times with DMSO to remove the residual crosslinker EGDGE and the unreacted curdlan; finally the S gel was immersed in 1000 mL of distilled water three times to remove the DMSO from the gel. The dry S gel was obtained by vacuum freeze-drying.

Curdlan hydrogel with a random coil skeleton (C gel) was prepared in 2% NaOH aqueous solution at 25 °C. The specific procedure is the same as that used for the S gel, except that the NaOH concentration is higher and the experiment is performed at room temperature.

### 2.3. Diffuse Reflectance Infrared Fourier Transform Spectroscopy

Diffuse reflectance infrared Fourier transform spectroscopy was performed on raw curdlan powder, S gel, and C gel using a Thermo Scientific (Waltham, MA, USA) Nicolet iS50R FIIR spectrometer. The sample compartment of the cell was filled with the as-prepared sample (~20 mg). Spectra were recorded from 4000 cm^−1^ to 650 cm^−1^, with 4 cm^−1^ resolution, and 120 scans were peformed in each acquisition.

### 2.4. X-ray Diffractometry Analysis

The structures of raw curdlan powder, S gel, and C gel were recorded by an X-ray diffractometer (X’Pert3 Power, Malvern Panalytical, Malvern, UK) using Cu Kα-radiation, with a step of 0.02° operating at a respective voltage of 40 kV and a current of 40 mA. MDI Jade 6.0 software was used to calculate the degrees of crystallinity of the S gel and the raw curdlan powder.

### 2.5. Scanning Electron Microscopy Analysis

The specimens of raw curdlan powder, lyophilized S gel, and C gel were fixed on a metal support and sputtered with a layer of Pt under vacuum. Next, the surface morphology was examined using a scanning electron microscope (SEM, itachiS4800, Hitachi, Japan). The magnification and accelerating voltage of SEM were 150× and 10 kV, respectively, and the working distance was adjusted in the range of 7.8–8.4 mm.

### 2.6. Swelling Experiments of S Gel and C Gel

Lyophilized S gel and C gel were placed in 1.5 mL centrifuge tubes, which were then filled with water. The temperature was controlled at 25 °C. One day later, the centrifuge tubes were centrifuged at 1K rpm, and the upper-layer liquid was wiped off using filter paper. The equilibrium swelling ratio of gel, m/m_0_, was obtained by measuring the weight of the swollen gel (m) and the weight of dry gel (m_0_).

### 2.7. NMR Experiments

A total of 5 wt% raw curdlan powder was dissolved in 0.4 wt% and 2 wt% NaOH aqueous solution, respectively, then loaded in 5 mm NMR tubes and measured using a Bruker 600 MHz Advance III spectrometer equipped with a BBO probe. Chopped small pieces of dry S gel or C gel (~3 mg) were first placed into a 4 mm HR-MAS zirconia rotor and weighed; then, 15 times the weight of water was added and the sample was sealed in the rotor for one day to maintain equilibrium swelling. The ^13^C high-resolution MAS (HRMAS) NMR experiments were performed on a Bruker 600 MHz Advance III spectrometer equipped with a 4 mm HR-MAS probe at spinning rate of 3 kHz. The temperature was controlled by a Bruker cooling unit with an accuracy of 0.1 °C. A 15 min equilibrium time was required for each NMR experiment to maintain the temperature balance.

### 2.8. Activation of Factor G from Tachypleus Amebocyte Lysate

The curdlan gels (S gel; 1 mg) were swollen in 1 mL of Tris•HCl buffer for 6 h at 37 °C, and then incubated with a series of tachypleus amebocyte lysate agents (5, 8, 16, 20, 24 μL), respectively. After 30 min, the azo reagent containing Boc-Leu-Gly-Arg-4-nitroanilide (pNA) was added. Finally, the absorbance of the UV–Vis spectra (HP 8453 spectrophotometer) at 545 nm was determined at room temperature. The absorbance of C gel (1 mg) with 20 μL of tachypleus amebocyte lysate agent was measured as a control.

## 3. Results

### 3.1. Structure Characterizations of S Gel

Recently, our group reported that curdlan chains adopt the unique single-helical conformation in low NaOH concentrations (0.04 wt%–0.60 wt%) and under suitable temperatures (35–50 °C) [20]. Utilizing this conformation property of curdlan in NaOH aqueous solution, we prepare a curdlan hydrogel with a single-helical skeleton (named S gel). Figure 1 displays the preparation scheme of S gel in 0.4 wt% NaOH at 40 °C and another curdlan hydrogel, with a random coil skeleton (named C gel), in 2 wt% NaOH at 25 °C using the biocompatible chemical crosslinker EGDGE. The corresponding diffuse reflectance infrared Fourier transform spectra for lyophilized S gel, C gel, and raw curdlan powder are shown in Figure 2a. When compared with that of the raw curdlan powder, the diffuse reflectance infrared Fourier transform spectra for lyophilized S gel and C gel display broad absorbance bands (i.e., OH stretch at 3480 cm^−1^, asymmetric CH stretch of CH2 at 2920 cm^−1^, and CO stretch and OH deformation at 1250 cm^−1^), indicating the occurrence of the crosslinking reaction of the curdlan chains with the crosslinker. It is widely accepted that the epoxy group can form ether bonds with the hydroxyl groups through the ring-opening reaction, which has been used in the crosslinking of polysaccharides [21,22]. EGDGE is a suitable reagent for biomedical application due to its amphiphilicity and easy removal from the human body. Curdlan contains three chemically nonequivalent hydroxyl groups: 6-OH, 4-OH, and 2-OH. It was reported that the reactivity of the OH groups in β-1,3-glucan with the sulphation reagent obeys the following order: 6-OH > 2-OH >> 4-OH [12,23]. The 6-OH is oriented toward the outside of the single-helix, displaying relative freedom. Our previous 2D COSY NMR data demonstrated that the 2-OH protrudes towards the center of the curdlan chain due to the strong O(2)^…^H-O(2′) intramolecular hydrogen-bond interaction, playing an important role in the rebuilding of the single-helical conformation [20]. Thus, the ethylene oxide groups of EGDGE should be introduced mainly at the 6-OH position for S gel in order to form the unique gel network structure with the single-helical skeleton, while for C gel, EGDGE should be introduced at all three OH positions, in the order of 6-OH > 2-OH >> 4-OH.

The X-ray diffraction pattern can provide the amorphous or crystalline properties of materials [24]. Previous studies demonstrated that the introduction of a crosslinker could change the structural and physicochemical characteristics of polysaccharides [25,26,27]. To evaluate the crystallinity of the two curdlan gels (S gel and C gel), the XRD diffraction diagrams of lyophilized S gel, C gel, and raw curdlan powder in the 2θ range of 5–60° are plotted in Figure 2b. Raw curdlan powder has three characteristic diffraction peaks at 2θ = 6.7°, 11°, and 20° [24,28]. The peak at 2θ = 6.7° corresponds to ordered triple-helical conformation occurring in lentinan, schizophyllan, and the annealed curdlan heating treatment at 120 °C [29]. The intense peaks at 2θ = 20° should be attributed to the single-helical crystal form of raw curdlan [28], and the degree of crystallinity of raw curdlan powder is calculated to be 31%. As for the S gel, the diffraction peaks at 6.7° and 11° disappear, and the diffraction peak at 20° is strong and slightly broader, revealing that S gel possesses no triple-helical conformation, maintaining its single-helical crystalline structure during the crosslinking process. The calculated degree of crystallinity of S gel is ~24%, which is slightly lower than that of raw curdlan powder. For C gel, the diffraction peak at 2θ = 20° is very weak, inferring that C gel has an amorphous structure. Previous reports confirmed that modification at the 6-OH group scarcely reduces the helix-forming and guest-binding functions of β-1,3-glucan [30,31], and thus, it is reasonable that S gel crosslinked at the 6-OH position can maintain a crystaline structure.

SEM is an effective tool for analyzing the surface morphology of native polysaccharides and their derivatives at a submicron level [32,33]. For comparison, the SEM images of raw curdlan powder, C gel, and S gel are displayed in Figure 3. The raw curdlan powder shows a sawdust-like image, while the surfaces of the C gel and S gel are continuous due to chemical crosslinking of the EGDGE. Specifically, C gel has a smooth surface, with ~30 μm pits and pores, revealing that the textural structure of C gel with a coiled skeleton is tightly packed. In contrast, S gel has a honeycomb-like surface, with large pores ranging from 50 to 200 μm, and the surface of the pore wall is smooth. The SEM images in Figure 3 provide direct evidence of the formation of S gel and C gel, and the different microstructures can reflect the swelling behaviors of prepared curdlan hydrogels.

### 3.2. Swelling Properties of S Gel and C Gel

The 1D ^13^C NMR spectrum of 5 wt% curdlan in 0.4 wt%NaOH aqueous solution at 40 °C, before the addition of a chemical crosslinker, is shown in Figure 4a, and the corresponding 1D ^13^C NMR spectrum of 5 wt% curdlan in 2 wt% NaOH aqueous solution at 25 °C is shown in Figure 4c. The carbon signals of curdlan are assigned according to those previously report [34]. The curdlan chain in 2 wt% NaOH aqueous solution exists in a coiled state, and all the carbon signals are very sharp, whereas in 0.4 wt% NaOH aqueous solution, the corresponding line widths of C1-C5 located on the pyranoid ring of curdlan are about four times wider than those in 2 wt% NaOH solution, and the line width of C6 hanging on the pyranoid ring of curdlan is less effective. The conformation-dependent C1 and C3 signals also shift downfield in 0.4 wt% NaOH aqueous solution. The changes in the ^13^C chemical shifts and the line widths of curdlan are consistent with our recent report [20], revealing that the conformation of the curdlan chain in 0.4 wt% NaOH solution at 40 °C is a single-helix.

Once the curdlan chains in the NaOH aqueous solutions are chemical crosslinked by EGDGE, the equilibrium swelling ratios of the obtained gels in H_2_O at room temperature are measured. The results are 18.5 ± 2 for S gel and 6.3 ± 2 for C gel, inferring the higher swelling ability of S gel than that of C gel. Combining the SEM results of S gel and C gel, the higher swelling ability of S gel should be responsible for its porous surface structure, as shown in Figure 3.

After chemical crosslinking, the corresponding 1D ^13^C HRMAS NMR spectra of swollen S gel and C gel with the 1:15 gel:water by mass are shown in Figure 4b,d. For C gel (Figure 4d), only the C6 peak is visible, inferring that the C gel network is less mobile. C gel can form a tight network through introducing EGDGE to the three OH positions, which should be responsible for the disappearance of C1-5. It is worth noting that all the ^13^C signals of S gel are shown in Figure 4b, except for the C4 signal overlapped by EGDGE. Although the ^13^C line widths of S gel are wider than those shown in Figure 4a, it can be confirmed that the network of S gel possesses high mobility. The high chain motion of the swollen S gel effectively averages out the ^1^H-^13^C dipole–dipole coupling, leading to the appearance of carbon resonances of the single-helical skeleton in S gel. However, the networks of C gel are tightly packed, and the ^1^H-^13^C dipole–dipole coupling becomes very strong, which dramatically broadens the ^13^C peaks, resulting in the loss of the pyranyl signals in C gel. EGDGE possesses two epoxy groups that can crosslink two curdlan chains. The single-helical conformation of the curdlan chain has a persistence length = 4.2 nm [20]; once EGDGE is grafted onto one of the single-helical curdlan chains, the pendulous epoxy group are spatially more difficult to attach to the other single-helical curdlan chain during the formation of S gel. Generally, the NMR signals of the crosslinking points are invisible in hydrogels. In Figure 4b, the linewidth of the EGDGE signals is about twice as wide as that of the free EGDGE, and thus, these EGDGE signals should belong to the pendulous EGDGE. As a note, the crosslinking efficiency of EGDGE is very low, thus requiring a large amount of excess. We are unable to give the accurate proportions of the bidirectional, monodirectional, or non-reactive EGDGE. From the DRIFTS, XRD, SEM and NMR results shown in Figure 2, Figure 3 and Figure 4, we confirm that a curdlan hydrogel with the single-helical skeleton is prepared, and this hydrogel maintains its fascinating crystaline structure, with high mobility in water at room temperature.

### 3.3. Actiation of Factor G from Tachypleus Amebocyte Lysate by S Gel

Conformation is a key factor determining the bioactivities of β-1,3-glucan. High-resolution, solid-state ^13^C NMR spectroscopy confirmed the necessity of a single-helical conformation for the Factor G-activating property of β-1,3-glucan [13]. Factor G from amebocyte lysate is thought to be very sensitive for the detection of the presence of the single-helical conformation of β-1,3-glucans [35]. Here, a chromogenic end-point tachypleus amebocyte lysate kit is used to explore the specific conformational–activity relationship of S gel. The binding of S gel with the α-subunit of G-factor from tachypleus amebocyte can convert the proclotting enzyme to the clotting enzyme that hydrolyzes the amido groups of Boc-Leu-Gly-Arg-4-nitroanilide (pNA) substrate to free pNA. Then, we record the absorbance of free pNA at 545 nm, which reflects the ability of S gel activating Factor G. In Figure 5a, the absorbance intensity at 545 nm is noticeably enhanced for S gel with an increasing tachypleus amebocyte lysate fraction. The corresponding activation vs. the tachypleus amebocyte lysate fraction curve is shown in Figure 5b. The absorbance is linear from 5 μL to 16 μL, and then becomes stable at 20 μL of tachypleus amebocyte lysate, meaning that the interaction of S gel with Factor G is saturated at 20 μL tachypleus amebocyte lysate per mg S gel, while the potency of activation of Factor G is negligible for C gel in Figure 5a. It was previously reported that the aniline blue dye is more reactive with the open single-helical conformation of NaOH-treated schizophyllan (SPG, branched β-1,3-glucan) than with its native triple-helical conformation in solution, and the effect of adding aniline blue is to decrease the overall responsiveness to limulus amebocyte lysate activity, confirming that dye and tachypleus amebocyte lysate are more responsive to a single-helical conformation of β-1,3-glucans [7,8]. Our prepared unique curdlan chain with the single-helical conformation in NaOH aqueous solution could also encapsulate Congo Red to form a stable, supramolecular dye assembly.

Treatments of β-1,3-glucans, such as the addition of a strong base or enhancing the temperature, are not stable and the β-1,3-glucans will undergo a slow renaturing process in solution after neutralization, i.e., the open single-helical conformation gradually reverts to the native triple-helical conformation. Young et al. found that about 84% of the activity of renatured SPG in a limulus amebocyte lysate test is lost within four days [7]. As for another β-1,3-glucan, grifolan, Nagi et al. revealed that about half of the NaOH-treated grifolan, with the single-helical conformation, gradually changes to the triple-helical conformation over a week at 4 °C [36]. The conformations of curdlan easily undergo transformation when in vivo and in vitro environments change, which significantly influences its bioactivity. Our prepared S gel can avoid the transformation of curdlan conformations and immobilize the single-helical skeleton within its highly swollen network. Surprisingly, S gel, with its unique of single-helical skeleton and minimal chemical modifications, shows a satisfactory activity to activate Factor G, which should be a valuable material in the food industry and the pharmaceutical field.

## 4. Conclusions

In nature, β-1,3-glucans are widely distributed in fungi, bacteria, plants, and algae, playing vital immunomodulatory, antitumor, and anti-inflammatory roles. Curdlan is the simplest β-1,3-glucan, possessing multi-biological activities and thermo-gelling properties. But the architecture of the curdlan is complex and environmentally variable, which hinders the understanding of the structure–activity relationship of curdlan. We prepare a chemically crosslinked curdlan hydrogel with a unique single-helical skeleton (S gel) and a curdlan hydrogel with a coiled skeleton (named C gel), which are confirmed using the DRIFTS method. Using the XRD method, it can be determined that S gel maintains the single-helical crystal structure during the crosslinking process, and the degree of crystallinity of S gel is ~24%, which is slightly lower than that of raw curdlan powder. Scanning electron microscopy confirms that S gel is a continuous network structure, with large pores measuring 50–200 μm, which is confirmed by its high swelling property in water. We also find that the carbon signals of the single-helical skeleton of the swollen S gel are visible, indicating that the single-helix skeleton of the S gels exhibits fascinating mobility at room temperature, observed through the ^13^C HRMAS NMR method. Utilizing a tachypleus amebocyte lysate kit, the ability of S gel to activate Factor G is confirmed. On the other hand, C gel has no crystal structure or activity. Our prepared S gel can avoid the transformation of the curdlan conformation and retain its biological activity to Factor G, which should identify it as valuable material for use in the food industry and the pharmaceutical field. Our work deepens the understanding of the relationship between the single-helical structure and the function of curdlan, promoting the development and application of β-1,3-glucans.

## Figures and Tables

**Figure 1 polymers-16-01323-f001:**
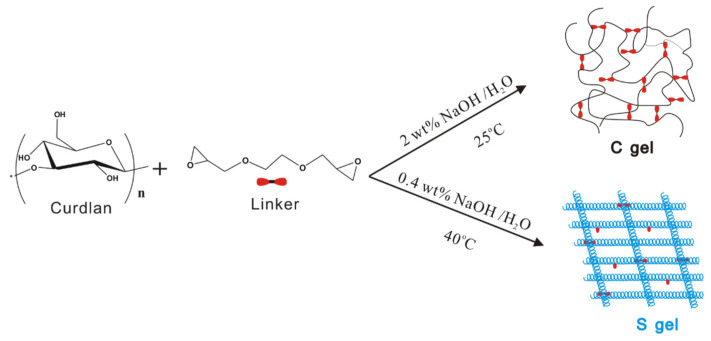
Preparation scheme of S gel and C gel.

**Figure 2 polymers-16-01323-f002:**
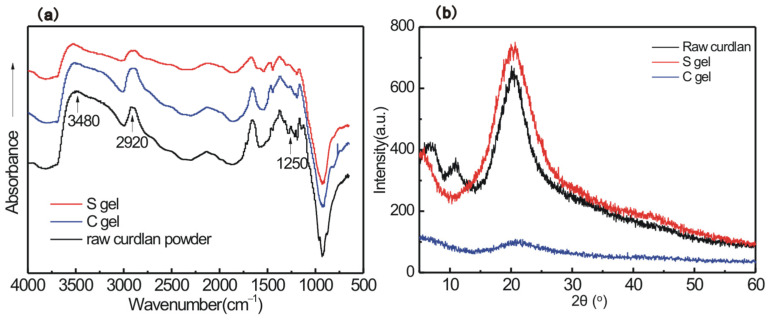
(**a**) DRIFTS spectra and (**b**) XRD curves of lyophilized S gel, C gel, and raw curdlan powder at room temperature.

**Figure 3 polymers-16-01323-f003:**
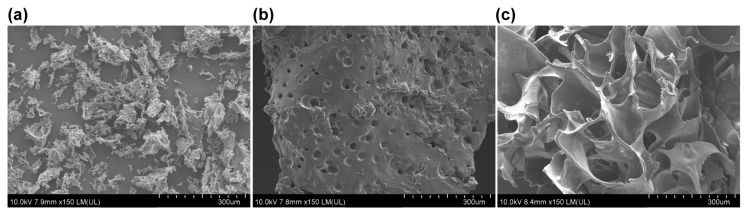
SEM images of lyophilized raw curdlan powder (**a**), C gel (**b**), and S gel (**c**) at room temperature.

**Figure 4 polymers-16-01323-f004:**
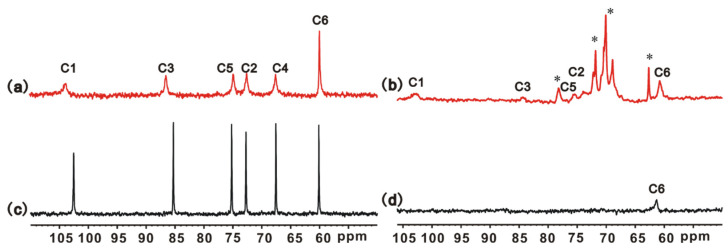
^13^C NMR spectra of 5 wt% curdlan/0.4 wt% NaOH aqueous solution at 40 °C (**a**), and 5 wt% curdlan/2 wt%NaOH solution at 25 °C (**c**); ^13^C HRMAS NMR spectra of swollen S gel (**b**), C gel (**d**) with the 1:15 gel:water by mass at 25 °C; and the asterisk (*) represents the peaks of the crosslinking agent.

**Figure 5 polymers-16-01323-f005:**
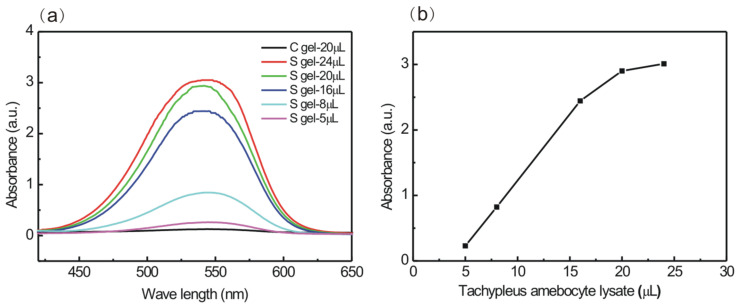
(**a**) UV–Vis curves of S gel and C gel activation of coagulation Factor G in a series of tachypleus amebocyte lysate agents (5, 8, 16, 20, and 24 μL), and (**b**) the curve of maximum absorbance of S gel vs. tachypleus amebocyte lysate fractions.

## Data Availability

The data are contained within the article.

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
