# Peer review of "Preparation of Single-Helical Curdlan Hydrogel and Its Activation with Coagulation Factor G"

_polymers, 2024, doi:10.3390/polym16101323_

Round 1
Reviewer 1 Report
Comments and Suggestions for Authors
The manuscript entitled "Preparation of single-helical curdlan hydrogels and its activation on coagulation Factor G" is an original work dealing with the conformation and activity of the beta-1,3-glucan curdlan. The work is overall interesting with potential scientific significance. The novelty of the prepared single-helical hydrogel is well pointed, but its relation to activity should be better described and discussed in comparison to previous findings. Furthermore, there are some issues of concern that would require the authors' attention. The numerical list of my comments is presented below:
1. The abstract is rather poorly written and very difficult to understand. I suggest the authors to rephrase it in order to present the work in more attractive way.
2. The introduction should provide more details into the effects of curdlan and a better background on choosing coagulation Factor G as a marker of the activity.
3. Was FTIR analysis performed to prove the crosslinking of the hydroxy groups?
4. How was the crystallinity determined? Please, add explanation in the methodology.
5. How was the ratio 1:15 for swelling determined? Were there some experiments regarding swelling ability performed? I suggest adding such information to the manuscript.
6. In figure 4 "a" and 'b" are missing even though they are referred to in the text body.
7. Please, elaborate on the statement that curdlan with single-helical skeleton "should be a valuable material in food industry and pharmaceutical field". The presented data is not clearly related to this statement.
8. Please, introduce all abbreviations the first time they appear in text (AFM, TEM, GPC, etc.)
Comments on the Quality of English LanguageThe English would need some moderate to extensive English editing. There are some very unclear sentences (e.g. "to make DMSO rapidly diffusing from the gel", "crystallinity of S gel is a little decrease", "widths of S gel become further broad compared...", etc. The are some very long sentences which make the comparison difficult to follow. Some typos can also be found.
Author Response
"Please see the attachment."

Reviewer 2 Report
Comments and Suggestions for Authors
The authors present powder XRD and 13C HRMAS NMR characterization of the single helical curdlan hydrogels prepared by mild treatment with alkali and heat. I would recommend publication of this article once the following fairly minor points are addressed.
- The degree of crosslinking with respect to EGDGE amount in C and S gels are not shown in the paper. How much of the EGDGE is reacted bidirectionally, monodirectionally or washed out? NMR of the supernatant DMSO in washing can reveal some answers for this, as well as swelling ratio studies of the final gel as a function of EGDGE concentration. Since the difference in crosslinking reaction (6-OH vs all OHs) is their claim for discrepancy in gel dynamics, I believe this part is integral to the main claims of the paper.
- Related, the authors claim that the 13C HRMAS NMR peaks in swollen S gels are more visible than C gels, correlating it to high mobility of the gel backbone. Why are the EGDGE peaks much sharper in panel (b) whereas they are absent in (d)? Does this indicate that for S gels, the chemical crosslinking is unsuccessful resulting in mono-pendant side chains?
Comments on the Quality of English Language
- Some grammatical and spelling errors are observed throughout the text (for example, line 16 "than that row powder" does not make sense). Copyediting is recommended.
Round 2
Reviewer 1 Report
Comments and Suggestions for Authors
The revised version of the manuscript "Preparation of single-helical curdlan hydrogels and its activation on coagulation factor G" has addresses most of my previous concerns. It was overall improved in terms of readability and presenting data. Nevertheless, I still consider it requires major revision before being suitable for publication. Below, I list my current questions:
1. It is very good that the authors have provided SEM micrographs of the different curdlan gels and raw material. In my opinion, they illustrate the morphology of the systems but cannot replace the FTIR analysis. When a crosslinking step is performed, it should be proven by suitable technique what kind of bonds are formed.
2. The magnification for SEM should be stated. Furthermore, the scale bars in figure 3 differ which makes the comparison inappropriate. Please, provide all details of the SEM (magnification, acceleration) and unify the scale bars.
3. How many grams of the lyophilized gels were introduced in the centrifuge tubes for the swelling studies?
4. It is still not clear why the ratio for NMR experiments was set to 1:15 gel:water. The swelling data suggests different amounts of water.
Comments on the Quality of English LanguageThere are a lot of grammar mistakes throughout the text - 3rd person singular for verbs is not used; singular and plural forms not properly addressed with verbs; verbs are missing in some sentences; inappropriate combinations of verbs with prepositions (e.g. possesses with) and others. Thorough English revision is highly advisable.
Round 3
Reviewer 1 Report
Comments and Suggestions for Authors
The second revision of the manuscript has provided sufficient information and adequate modifications. I consider the work presentation has been significantly improved and the overall merit is now sufficient for being suitable for publication.